Charge and hydrophobicity are key features in sequence-trained machine learning models for predicting the biophysical properties of clinical-stage antibodies

Hebditch Max max.hebditch@manchester.ac.uk
Warwicker Jim jim.warwicker@manchester.ac.uk
School of Chemistry, Manchester Institute of Biotechnology, University of Manchester , Manchester , United Kingdom
Paci Emanuele
Electronic publication date: 2019 Dec 18
Publication date: 2019
Volume: 7
Electronic Location ID: e8199
Received 2019 Aug 28; Accepted 2019 Nov 13
Copyright: ©2019 Hebditch and Warwicker
Copyright year: 2019
Copyright holder: Hebditch and Warwicker
License: This is an open access article distributed under the terms of the Creative Commons Attribution License, which permits unrestricted use, distribution, reproduction and adaptation in any medium and for any purpose provided that it is properly attributed. For attribution, the original author(s), title, publication source (PeerJ) and either DOI or URL of the article must be cited.
License URL: https://creativecommons.org/licenses/by/4.0/

Keywords: Machine learning, Antibodies, Biotherapeutics, Bioinformatics, Biophysics

Funding: UK EPSRC EP/N024796/1 This work was supported by the UK EPSRC (grant EP/N024796/1). The funders had no role in study design, data collection and analysis, decision to publish, or preparation of the manuscript.

==============================
Improved understanding of properties that mediate protein solubility and resistance to aggregation are important for developing biopharmaceuticals, and more generally in biotechnology and synthetic biology. Recent acquisition of large datasets for antibody biophysical properties enables the search for predictive models. In this report, machine learning methods are used to derive models for 12 biophysical properties. A physicochemical perspective is maintained in analysing the models, leading to the observation that models cluster largely according to charge (cross-interaction measurements) and hydrophobicity (self-interaction methods). These two properties also overlap in some cases, for example in a new interpretation of variation in hydrophobic interaction chromatography. Since the models are developed from differences of antibody variable loops, the next stage is to extend models to more diverse protein sets.

Availability

The web application for the sequence-based algorithms are available on the protein-sol webserver, at https://protein-sol.manchester.ac.uk/abpred, with models and virtualisation software available at https://protein-sol.manchester.ac.uk/software.

Introduction

The promise of therapeutic monoclonal antibodies relies on the ability of the pharmaceutical industry to develop large scale manufacturing processes that can produce safe, reproducible, and economical formulations. Identifying problematic antibody formulations, as early as possible in the drug discovery programme, has become a key area of research. To serve this interest, researchers have identified various experimental platforms, and developed theoretical tools, in an attempt to identify antibodies that may exhibit deleterious solution properties, also referred to as developability issues (Jarasch et al., 2015; Kohli et al., 2015). The use of experimental methods necessitate the production of a large number of candidates, which is both expensive and time consuming. There is also the cost of conducting the biophysical characterisation assays and interpreting the result. For these reasons, there has been interest in developing new techniques to minimise sample requirements, or increase throughput (Razinkov, Treuheit & Becker, 2013; Man et al., 2019). To help alleviate the sample requirement issues for experimental methods, several groups have developed theoretical tools to assay the solubility, or developability, before any expression or purification is required (Lauer et al., 2012; Obrezanova et al., 2015; Hou et al., 2018; Sankar et al., 2018). Although excipients and solution conditions have a large effect on biophysical solution behaviour (Kamerzell et al., 2011; Ohtake, Kita & Arakawa, 2011; Lilyestrom, Shire & Scherer, 2012), the properties of the formulation will be determined de novo by sequence and structure, and thus form the basis for many theoretical approaches. There are a number of sequence-based predictors of protein aggregation, particularly as applied to amyloid proteins, in the literature (Tartaglia & Vendruscolo, 2008; Conchillo-Solé et al., 2007; Walsh et al., 2014), as well as more general antibody specific homology models (Marcatili et al., 2014; Leem et al., 2016; Weitzner et al., 2017), and recent work has applied these techniques for predicting the solubility of biotherapeutics (Sormanni et al., 2017; Raybould et al., 2019).

The use of these in silico candidate screening techniques accelerates the biotherapeutic development process, through the identification of high value leads and new engineering targets (Shan et al., 2018), and in some cases even improving biological activity (Kumar et al., 2018), However, the development of these tools is reliant on the availability of high quality experimental datasets and is thus heavily dependent on the progress of experimental techniques. Notably, the recent release of antibody biophysical characterisation datasets (Goyon et al., 2017; Jain et al., 2017a) has allowed the development of further theoretical tools to predict, assess and understand the physicochemical properties that are correlated with the successful development of a therapeutic antibody, on a scale previously unattainable to academic researchers. The Jain et al. (2017a) report in particular is an excellent resource as it analysed 137 antibodies, representing a wide variety of late stage clinical therapeutics, across 12 different biophysical characterisation platforms. The study identified where there is overlap between complementary approaches and which platforms should be prioritised for assaying candidate therapeutic mAbs.

When available, previous work from our group has used experimental data to produce algorithms for both prediction and theoretical calculation which we have made freely and openly available as web applications on the protein-sol web server. Prior to the release of the high throughput biotherapeutic datasets, we have focussed on using other large datasets, such as the Niwa et al. (2009) E. coli solubility dataset, as a proxy for therapeutic proteins, to study the role of sequence information in predicting protein solubility (Hebditch et al., 2017). Using the Goyon et al. (2017) dataset we studied the importance of CDR (complementarity-determining regions) length and aromatic content for predicting behaviour on HIC (hydrophobic interaction chromatography) (Hebditch et al., 2018). Lastly, we have developed tools for predicting the presence of hydrophobic and charged patches as well as fold state stability (Hebditch & Warwicker, 2019) from crystal structures available in the PDB (Berman et al., 2007) and applied these observations to experimental work (Austerberry et al., 2017). After the release of the Jain et al. (2017a) dataset, reports have appeared in the literature using the dataset. For example, predictive models of HIC performance using QSPR models (Jetha et al., 2018) and a combined sequence and structure approach (Jain et al., 2017b). CDR properties of the Jain et al. (2017a) dataset have also been implicated in identifying antibodies with developmental issues (Raybould et al., 2019), and the dataset has also been used to benchmark aggregation prediction algorithms (Sankar et al., 2018).

In this report, we describe our approach to using machine learning algorithms trained on the Jain et al. (2017a) dataset. Models for all 12 biophysical measurement platforms are produced, with varying efficacies. Compared to other approaches, our models rely simply on sequence information which is readily available in comparison to structural approaches. To our knowledge, this report gives the first set of sequence trained models for predicting the performance on biophysical characterisation platforms important for assessing the developability of biotherapeutic antibodies. The models are interrogated for which sequence features contribute most significantly for each measurement, and clustering the models according to the relative importance of the sequence features is largely in accord with clustering from the experimental report (Jain et al., 2017a). From our analysis of sequence information, we associate charge and hydrophobicity calculated from amino acid propensity as the features of most importance. In a novel interpretation of results for HIC, a complexity is revealed whereby charge effects are hypothesised to be minor at low retention times, but major at high retention times, owing to the ionic strength gradient that is used to modulate hydrophobicity.

Methods

Dataset

The Fv (concatenated VH and VL) sequences for the 137 antibodies (mAb137), as well as the experimental result data for the 12 biophysical platforms were obtained from Jain et al. (2017a). The experimental methods were: AC-SINS (affinity-capture self-interaction nanoparticle spectroscopy), CSI-BLI (clone self-interaction by bio-layer interferometry), PSR (poly-specificity reagent), BVP-ELISA (baculovirus particle ELISA), CIC (cross-interaction chromatography), ELISA (enzyme-linked immunosorbent assay), HEK (HEK cell expression titer), HIC (hydrophobic interaction chromatography), SGAC-SINS (salt-gradient affinity-capture self-interaction nano-particle spectroscopy), SMAC (stand-up monolayer adsorption chromatography), SEC (size-exclusion chromatography), DSF (differential scanning fluorimetry).

Identifying explanatory variables

Following on from previous work where we used protein sequence features to estimate solution behaviour (Hebditch et al., 2017), we have used the same 35 sequence features in an attempt to understand the variance in the 137 antibodies with the 12 different biophysical characterisation assays. The 35 features are composed of the standard 20 amino acid propensities, followed by 7 amino acid composite scores (KmR = K-R, DmE = D-E, KpR = K+R, DpE = D+E, PmN = K+R-D-E, PpN = K+R+D+E, aro = F+W+Y) and a further 8 sequence features, fld = folding propensity (Uversky, Gillespie & Fink, 2000), dis = disorder propensity (Linding et al., 2003), bet = beta strand propensities (Costantini, Colonna & Facchiano, 2006), mem = Kyte-Doolittle hydropathy (Kyte & Doolittle, 1982), pI, absolute charge (calculated from the presence of amino acids with titratable moieties: K and R positive, D and E negative) and sequence entropy (Hebditch et al., 2017), reflecting different calculations of charge and hydrophobicity as well as previously established calculations used in the literature. The complementary determining regions (CDRs) were identified for each Fv sequence using a set of sequence based rules (Abhinandan & Martin, 2008).

Feature engineering and preprocessing the experimental datasets

For each experimental method, we first determined which features should be selected as explanatory variables. If we desired to simply maximise the R2 value, we could retain all of the variables as the R2 value will always increase with higher degrees of freedom (Kvålseth, 1985). However in order to generate the most robust, and interpretable, model it is preferable to reduce the number of variables used. For example, collinearity occurs in multivariate regressions when input/explanatory variables are correlated, and this correlation can destabilise the estimation of individual coefficients (Farrar & Glauber, 1967). If the VIF value is high, the variance of the coefficient in the multivariate model is high, and thus the estimation of the standard error is high. To account for this, for each experimental platform we selectively removed variables with a high VIF score (traditionally considered 10 and above) to address the issue of multicollinearity (O’brien, 2007). This list of non-collinear variables formed our first set of explanatory variables: VIF all. To identify the most important coefficients we conducted a mixed stepwise selection regression (Venables & Ripley, 2013) to minimise the Akaike information criterion (Akaike, 1998). This process resulted in the second of our two sets of explanatory variables for each biophysical platform: VIF selected. For each experimental dataset, we then scaled the explanatory variables to ensure that the coefficient value of each explanatory variable would be comparable, as well as for aiding the prediction of the statistical techniques.

Mathematical transformation of the experimental data

Many machine learning algorithms perform best on normally distributed datasets (James et al., 2013), and it is common practise to mathematically transform non-normal distributions in order to improve the predictive power of machine learning approaches. We noted that many of the values are in fact better described by a generalized extreme value distribution type 1 (Gumbel distribution). As many of the experimental distributions appeared to be significantly non-normal, we normalised all of the distributions using a mathematical transformation (see Table 1). For each dataset, we attempted to normalise the distribution of experimental values using the R package bestNormalize and then trained the algorithm against both the standard and normalised datasets using both the stepwise selected and complete sets of coefficients (Peterson & Cavanaugh, 2019). For the datasets with significantly non-normal distributions, the machine learning algorithm was then trained on these transformed experimental values and will thus produce regressions in the context of the transformed space. Although these transformed predictions do not have any physical meaning, they are still mathematically related to the original experimental value, and can therefore be used to compare between proteins in the mAb137 dataset.

Table 1 Chosen machine learning algorithm summary.

For each experimental dataset we tested a number of different algorithm, variable and mathematical transformation types.

	HIC	SMAC	CIC	ACSINS	ELISA	BVP	SGAC-SINS	PSR	HEK	DSF	CSI	ACC-STAB	
Algorithm	Elastic net	Elastic net	SVM	Elastic net	Random forest	Random forest	SVM	SVM	SVM	SVM	SVM	SVM	
Variables	VIF selected	VIF selected	VIF all	VIF selected	VIF all	VIF all	VIF all	VIF all	VIF all	VIF all	VIF all	VIF all	
Transformation	None	Ordered quantile	Ordered quantile	Ordered quantile	None	None	None	None	None	None	Ordered quantile	None	
R2	0.391	0.353	0.306	0.268	0.383	0.355	0.215	0.316	0.1121	0.13	0.169	0.086	
p-value	2.33E–17	7.33E–15	4.46E–17	6.46E–14	4.95E–77	6.85E–68	2.30E–39	2.39E–10	1.87E–09	4.49E–08	1.24E–05	2.82E–01	

Statistical model selection and cross-validation

We tested the performance of both parametric and non-parametric machine learning algorithms. The advantage of non-parametric methods is that they are generally unbiased as they do not expect the data to fit to an a priori approximation (James et al., 2013). This flexibility however comes at the cost of generally requiring larger datasets in order to model the relationship, and the increased degrees of freedom can lead to overfitting the data. Parametric algorithms are easier to interpret and are more useful for inferential statistical approaches, but are however more likely to be biased as they assume a structure to the data that may not exist (James et al., 2013). To determine which algorithm should be used for each experimental method, we tested 11 different regression algorithms representing a both parametric and non-parametric algorithms. By using a broad range of different machine learning theoretical approaches, we are unbiased in our model selection as we make no assumptions about the structure of the data. Each algorithm was provided with both sets of explanatory variables (VIF all and VIF selected) and the normalised and standard datasets for each of the 12 experimental methods.

For predicting how machine learning models perform on unseen data a validation approach is required. Traditionally a hold-out, or lock box, validation approach is favoured (Chicco, 2017), however for a dataset of this size (n = 137), a hold out-approach for estimating model performance would be problematic. Firstly it would be difficult to ensure that both the validation and hold-out would be truly representative of the sample, for this reason, any partition of the data would be highly variable due to randomness in selecting the hold-out set. Secondly, machine learning approaches perform worse with fewer observations, and by necessity a hold-out validation approach will immediately remove a substantial portion of the data for validation. For these reasons, we chose to use cross-validation for estimating the test error directly from the training data, a technique common in life sciences (Krstajic et al., 2014). A traditional approach to cross-validation is the k-fold technique where the data is divided into k folds, with 1 of the folds being put aside for validating a model trained on the remaining k-1 folds. Compared to the hold-out validation method, this ensures that we can use the entirety of our data for training the algorithms whilst still retaining an estimate of performance on future data. Other cross-validation approaches to studying the mAb137 dataset (Jetha et al., 2018) have used leave-one-out cross-validation, which is a special subset of k-fold cross-validation where the number of folds is equal to the number of observations, and thus each validation fold consists of a single observation with the remaining folds used to train the algorithm. The leave-one-out form of cross-validation will tend to have higher variance than a k-fold of 10 approach due to the high similarity of the training sets, which only differ by 1 sample (Kohavi, 1995). This means that each dataset in the leave-one-out approach is highly correlated, whereas if a smaller number of k-folds are used, the training sets are more diverse and should therefore provide a more accurate estimate of the test error as a proxy for performance on unseen data due to the bias–variance trade-off (James et al., 2013). As we are most concerned with providing a robust predictive algorithm which can be applied to future unseen data, we chose to use a 50 times repeated 10-fold cross validation approach as a trade-off between providing the model with as much training data as possible, whilst maintaining a robust and diverse training set to avoid over fitting (Braga-Neto & Dougherty, 2004; Krstajic et al., 2014). For the HIC dataset, we excluded the experimental values with a value of 25, as these were arbitrarily assigned a value of 25 due to exceeding the maximum measurement time and were thus misleading for the training. Both the algorithms and cross-validation were implemented using the caret (Kuhn, 2008) package in R version 3.5.1.

Selection of machine learning algorithms and data input

After training each algorithm on the original untransformed and transformed experimental data, the algorithms with the lowest mean average error (MAE) were chosen for further exploration. For most of the experimental datasets, the best performing algorithms were the elastic net, a linear algorithm (Zou & Hastie, 2005), using the stepwise selected variables, and the non-linear algorithms: support vector machines (SVM) (Drucker et al., 1997), and random forest (Ho, 1995), both using the complete set of non-collinear variables (see Table 1). The selected model for each experimental method was then used to predict the entire experimental dataset to obtain predicted values for each of the proteins.

Meta score

We also provide a meta score which combines and averages multiple biophysical platforms. The meta score is calculated by ranking the original Jain dataset in order from best to worst result, and then calculating where the candidate sequence falls within that ranking for each biophysical platform. We rescale the rankings from 1–100, with 1 being predicted to be the best, and 100 predicted to be the worst, with the rankings ordered dependent on whether higher or lower better values are preferable. We then combine and average the ranks for the biophysical platform. For META X we average the rankings for ELISA, BVP, PSR, CSI, ACC STAB and CIC, and for META Y we average SMAC and HIC. The lower the ranking the better for each group, and thus the closer to origin (0,0) the better we predict the candidate to behave on average across the platforms.

Results

Web application and model availability

Previous work from our group has focussed on developing predictive models (Hebditch et al., 2017) and theoretical tools (Hebditch & Warwicker, 2019) which we have made freely available as a suite of web-tools for the wider research community at https://protein-sol.manchester.ac.uk. Accordingly, we have made all twelve machine learning algorithms available at protein-sol. The user can enter a candidate Fv sequence into the web application, which is then processed using the same methodology as described in this study. The sequence composition scores of the new sequence are preprocessed for scale, and where applicable mathematical transformations applied (Table 1). The composition variables are then used as new inputs for the trained algorithms to obtain predictions for the 12 biophysical experiments. The web application provides an interactive scatter plot, with the original, or transformed, experimental value on the x-axis, and the predicted value from the machine learning algorithm for the same protein on the y-axis (Fig. 1). As the new candidate sequence has only a predicted value, we assign x = y. The web application presents the 12 predicted experimental values and the calculated META value in the context of the original mAb137 dataset to allow the user to assess the prediction. Hovering over the individual points on the scatter graph provides the name and FDA approval stage of the protein in question, as well as the original experimental value and the predicted value from the machine learning algorithm for comparison to the candidate protein. Where the machine learning algorithm has been trained on a mathematically transformed dataset (see ‘Mathematical transformation of the experimental data’) it is important to note that the values on the x and y axis are reported on the same transformed scale. For the candidate sequence the user is also given a ranking, scaled from 1 − 100 where 1 is always the preferential ranking, for each experiment, allowing the user to contextualise how the candidate sequence performs in comparison to the mAb137 set of clinical stage therapeutics. The heat map is colour coded for each Fv dependent on the threshold value. We use threshold values, available for 10/12 experimental platforms, from the original Jain et al. (2017a) study, for the remaining 2/12 (HEK and DSF) we set the threshold to mark values that rank within the worst 10% of the experimental values. If the predicted value is above the threshold value for the experiment, the corresponding square is coloured red, otherwise it is coloured green. Hovering over the heat map changes the displayed scatter graph to display the predictions for that category, as well as the ranking of the candidate sequence for that experiment.

Figure 1 Demonstration of the protein-sol web application.

The x axis of the scatter plot is the original experimental value, or mathematical transformation thereof, and the y axis is the prediction for that protein. The protein submitted by the user is coloured orange, and the mAb137 dataset is coloured green with the three shades representing the FDA approval stage. The heat map below is coloured red if the candidate protein is predicted to lie beyond the threshold and green otherwise.

The web application allows users to easily visualise and understand the predictions for single Fv sequences. If the user wishes to make predictions for multiple proteins, or implement the abpred software into their own pipelines, we are also providing the complete suite of software both as a repository with instructions for installing dependencies, and also as a docker image which is an industry standard form of operating system virtualisation allowing the user to download a preconfigured image containing all of the required software designed to run cross platform on Linux, Windows and macOS. These resources are available at https://protein-sol.manchester.ac.uk/software.

Models for 12 biophysical properties that characterise antibody behaviour

Sequence composition scores for the 137 Fv (combined VH and VL sequences averaged by length) from the Jain et al. (2017a) dataset (mAb137) were used to train multiple different machine learning algorithms, on the original, and mathematically transformed datasets. To ascertain generalisability, each algorithm was trained using 50-repeat 10-fold validation. From the cross-validation we obtained the MAE value which was used to choose determine which combination of algorithm and experimental data transformation best described each of the 12 experimental datasets (see methods). Finally, we then used the cross validated models to describe the entirety of the experimental data in order to obtain predicted values corresponding to each experimental value to power the prediction matrix (Fig. 2, Figs. S1 and S2). Using MAE as a qualification metric for model quality, we demonstrate that machine learning models trained simply on Fv sequence information, can provide reasonably accurate predictions for some of the 12 biophysical techniques, although accuracy varies substantially (Table 1). Although the models are selected in order to maximise the MAE rather than the correlation coefficient R2, it closely follows the model accuracy with a correspondingly low p-value. Our HIC model has an R2 value of 0.391, comparable to values in the literature. The Jetha et al. (2018) report describes three sequence based models with R2 values of 0.17, 0.1 and 0.23, and two more complicated structural based QSPR models with R2 values of 0.33 and 0.38. The Jain et al. (2017b) uses a combined sequence and structural approach, but reports an AUC of 0.87 rather than an R2 value making direct comparison difficult. Compared to both previous reports in the literature, our report provides the user with predictions based purely on amino acid sequence rather than the more complicated structural based approaches used by Jetha et al. (2018) and Jain et al. (2017b).

Figure 2 The scatter graph demonstrates the predictive power of the HIC model, where the original experimental value is on the x axis, and the prediction is on the y axis.

The closer each data point is to the y = x line the better the prediction. Predictions for HIC are generally close to the y = x line at lower, but not at higher retention times, suggesting that sequence based prediction is less reliable at higher HIC values.

Figure 2 shows one of the better performing models (HIC) with good agreement at lower at lower HIC values but less effective prediction at higher HIC values. The feature selection stage of the machine learning methods gives an indication of the sequence-based features that are most associated with particular biophysical properties. A complementary approach, is to examine correlations between sequence features and measured properties.

For each experimental dataset, we calculated the Pearson correlation coefficient between each calculated sequence feature and the experimental value for the entire Fv chain (Fig. 3). Inspection reveals that some biophysical measurements are associated with sets of sequence correlations that are of larger magnitude than for other measurements. These largely reflect our observations for models developed with machine learning (see Fig. S3 for a comparison of selected features and their correlation to the experimental result). Generally, the models shown in Fig. S1, with greater overall correlation values, and giving predictions that lie close to the y = x diagonal, are associated with greater correlation magnitudes reading across in Fig. 3. Consideration of sequence features that underlie models is important in further our understanding of molecular behaviour, as demonstrated with presentation of a new model for mAb behaviour in HIC, in a subsequent section.

Figure 3 Heat map of the Pearson correlation coefficient between the Fv sequence composition scores used in the abpred algorithms and the score on each of the experimental datasets for the mAb137 dataset.

Dark red values indicate a stronger positive correlation, and dark blue values indicate a stronger negative correlation.

Clustering of biophysical characterisations

Hierarchical clustering of biophysical characterisation for the 137 mAbs revealed 5 clusters (Jain et al., 2017a), which we are able to now associate with enrichment for higher correlations with certain sequence features. A grouping of positive correlations for charge-associated properties is apparent (Fig. 3) for the largest cluster identified previously (PSR, CSI, ACSINS, CIC), but we would add in a second of the original clusters (ELISA, BVP) that sits next to the largest cluster in the hierarchical tree. These 6 properties lie at the bottom of the heat maps in Fig. 3 and (to varying degrees) give correlations for absolute charge and negative charge subtracted from positive charge (PmN) i.e., overall net positive charge. The 6 assays in this cluster assess cross-interaction (BVP, ELISA, CIC, PSR) and self-interaction (CSI, ACSINS). We predict that, for the cross-interactions, negatively-charged proteins (or regions of proteins, and for some assays perhaps additionally phospholipids) are being targeted by more positively-charged CDRs in the mAbs. For self-interactions, absolute charge favours interaction, but it is less clear that this is a positive charge.

Whereas SMAC, HIC and SGAC-SINS were clustered according to biophysical characterisation (Jain et al., 2017a), Fig. 3 indicates (according to sequence properties) that SMAC and HIC are more closely related to each other than to SGAC-SINS, including positive correlation of aromatic content with association to the hydrophobic medium. This observation is consistent with our earlier modelling (Hebditch & Warwicker, 2019) for HIC measurements with a smaller dataset (Goyon et al., 2017), given the multiple dependencies of biophysical properties on sequence features it is unsurprising that models constructed with machine learning methods give correlations that are useful, but far from precise. It is likely that consideration of 3D structure can improve modelling when we have sufficient understanding of properties such as a shape-dependence of the hydrophobic effect (Hebditch & Warwicker, 2019), but 3D structure will not always be available and models are liable to error. In this context we have used the clustering of biophysical characterisation methods, which largely agrees when clustering is based on the measurements themselves (Jain et al., 2017a), to generate two combinations for prediction. Further, we loosely associate these two predictors with variation within a dataset of two overriding features. For cross-interaction and self-interaction, BVP, ELISA, CIC, PSR, CSI and ACSINS predicted rankings are averaged and displayed along the horizontal axis, and for hydrophobic interaction the rankings from HIC and SMAC are averaged and displayed on the vertical axis. The background to the plot, labelled ‘meta’ prediction, are the combined rankings calculated for the 137 mAb set. There is some cross-over between these two combinations, for example the charge effects in HIC (Fig. 3), and a positive correlation between CIC data and aromatic content, which is accommodated in the models for individual biophysical properties. In general terms though, this ‘meta’ prediction can be thought of as displaying variation associated with charge-related properties (horizontally), and variation associated with hydrophobicity (vertically).

HIC and the interplay between charge and hydrophobicity

For HIC we note that (Fig. 3) strong correlations between HIC value and aromatic content (positive) and absolute charge (negative). From Fig. S4, it is clear that there is no simple delineation between the high and low charge sequences (Fig. S4A) when considering the relationship between aromatic content and HIC, and between the high and low aromatic content sequences when considering the relationship between absolute charge and HIC (Fig. S4B). This suggests that the relationship between HIC and charge/hydrophobicity is not linear, and will thus not be captured by traditional linear models. To account for this, we calculated an interaction model between aromatic content and absolute charge for predicting HIC retention time (Fig. 4). Tracking the plot across at constant charge, about one third up along the charge axis, the expected behaviour is evident, where aromatic content is used to represent hydrophobicity. More generally though the plot shows that increased charge leads to lower retention times in HIC, for a uniform content of aromatic residues, evidenced by a worsening of the correlation between measurement and prediction at higher HIC values (Fig. 2). Deconvoluting and keeping track of features that are included in a model permits physical interpretation and re-evaluation that may be valuable for research into HIC methodology.

Figure 4 also includes a physical interpretation of these data. At higher ionic strength (relating to lower retention times in HIC), charge interactions between bound proteins are screened and thus high net charge proteins (Q) behave in a similar manner to those with low net charge (q). We suggest, at lower ionic strength (and longer elution times), charge interactions are no longer screened as effectively, so that repulsion between proteins with higher net charge (Q) would lead to greater elution of these proteins relative to those with smaller net charge (q). This combination of experimental data, informatics, and physicochemical analysis, leads to a novel interpretation of the complexity required in accounting for HIC data.

Discussion

Our models have been developed with specific biophysical characterisations and a single type of protein therapeutic, giving rise to the question of whether they retain predictive ability when either of these factors are changed. We have focussed on characterisation by HIC, largely since careful comparison of the data yields the insight that ionic strength variation during elution leads to a dependence on charge as well as hydrophobicity. Of the reported biophysical methods (Jain et al., 2017a), HIC is widely used, however it is applied in variants of the format that lead to altered ranges of measured retention times. A set of 97 mAb variants, targeting integrin α11, have HIC elution times in the range of 20–30 min, compared with those used for the current models that are centred around 10 min (Jetha et al., 2018). Models can still be assessed with correlation, or relative ranking, even when the measurement domains are different. A scatter plot of HIC values calculated with our model against experimental values has a similar appearance to a that produced with a sequence-based prediction developed in the original study (Jetha et al., 2018). The correlation coefficient for the scatter plot in our calculation is 0.35, lower than reported by Jetha et al. (2018) (0.46), but significant (p = 0.00044, 97 data points). The mAb variants in this study consist of mutations designed to reduce hydrophobicity, with varying degrees of success in the design, which are largely reflected in the scatter plot for model versus experimental HIC. Interestingly, a group of mutations based on Y30 are amongst the more poorly predicted set, and this residue is part of a relatively small hydrophobic patch and likely not involved in binding to integrin α11. In contrast the sequence space of the 137 mAb set will be determined mostly by altered binding to the various targets, and therefore qualitatively different to the 97 mAb set. Jetha et al. (2018) observe that structure-based modelling can be used to distinguish surface environments, consistent with our own report that an improved understanding of the shape-dependence of the hydrophobic effect is needed (Hebditch & Warwicker, 2019).

Figure 4 Plot of an interaction model trained on the aromatic content and absolute charge (from number of amino acids in the sequence with titratable moieties) for the mAb137 dataset and HIC experimental data.

Red indicates areas of higher HIC retention time, and blue areas of lower retention time. The schematic provides a suggested physical explanation for differences in HIC values in sequences with high aromatic content denoted by green non-polar interactions.

With regard to measurements for non-mAb systems, we follow Jetha et al. (2018) in comparing predicted HIC values with measured inclusion body (IB) percentage formation for 31 adnectin loop variants (Trainor et al., 2016), i.e., assuming that hydrophobicity is a contributing factor in IB formation. Our model for HIC correlates with the IB percentage data, yielding R = 0.668 (p = 0.00004, 31 data points).

These tests, against other mAbs and another loop-based protein affinity system, give us confidence that the models can be used for relative ranking of candidate molecules. There are clearly insufficiencies in the models, and correlations vary across models. Developments may come from inclusion of 3-dimensional structure (where available). Careful analysis of models can yield areas where improved physicochemical understanding is possible, illustrated here with HIC data.

Conclusions

A set of predictive models are presented for the 12 biophysical properties assessed in a landmark study of 137 mAbs (Jain et al., 2017a). The models have been developed from sequences of the (heavy and light chain) variable domains, using the 20 amino acid compositions and 15 sequence-derived features that represent physicochemical properties (Hebditch et al., 2017), and variation in those properties between the CDRs of the 137 mAbs. Machine learning methods have been used to access fits to the data that would be missed by linear models. For many of the experimental platforms, it turns out that linear models can account for much, but not all, of the variation observed, whilst for some of the measured properties it is difficult to obtain an effective predictive model. These deficits, for example in DSF (Tm) and HEK (expression titer) may highlight where sequence fails to capture salient structural features (Jetha et al., 2018; Raybould et al., 2019), or important factors in the solution environment. For example, structural stability of a mAb, either globally or for local regions, is likely to be correlated with Tm measurements, although this may be difficult to obtain accurately in a predictive model. Sequence-based prediction, though, gives effective models for many of the biophysical platforms and is accessible to users without structural information, it negates the requirement for comparative modelling (with its potential errors), and in prior work we find that 3D-based methods are still in development in regard to assessment of hydrophobic interactions at CDRs (Hebditch et al., 2018). An advantage of our methodology, with models for 12 biophysical properties, is that models can be clustered and examined in the context of common sets of sequence features with higher correlations. This clustering is similar to that established in the original report of the mAb137 set data. Further, the method allows new interpretation of underlying physicochemical behaviour. The example given is for HIC, where despite delivering one of the better models, we find that charge combines with hydrophobicity in a way that is difficult to capture precisely. However, the fundamental nature of this combination can plausibly be related to the HIC method. An ammonium sulphate gradient (high to low) is used to modulate hydrophobicity. We hypothesise that at shorter elution times, charge plays less of a role (with electrostatic interactions screened) but is more important at longer elution times, at lower ionic strength and with proteins with higher net charge repelling each other on the support. The current work complements other modelling studies, but of individual properties, built on the mAb137 set (HIC, Jain et al. (2017a); CIC, Kizhedath, Karlberg & Glassey (2019). It also adds to studies prior to reporting of the mAb137 set, that identified QSAR as an effective area for prediction of mAb solution properties (Sharma et al., 2014; Robinson et al., 2017).

A key question for prediction methods developed with data for mAbs, is how well they transfer to other proteins, particularly since the emphasis is on differences in CDRs. Since the detail of experimental procedures is likely to vary, the first step to other systems is to swap from a comparison of absolute values to rankings or correlations, using the numerical values given in our method. Following this procedure, our HIC prediction model is effective for another set of mAbs (Jetha et al., 2018) and a set of adnectin variants (Trainor et al., 2016). Both of these additional sets are centred on variation in defined loop regions. In the next stages of the work, wider variation in proteins will be studied, requiring collaboration with experimental determination to both extend the range of measurements, and to narrow the range of biophysical techniques to focus studies.

Supplemental Information

Supplemental Information 1 Better performing models

The scatter graphs demonstrate the predictive power of each model, where the original experimental value is on the x axis, and the prediction is on the y axis. The closer each data point is to y = x the better the prediction. For example, in Figure A, the predictions for HIC are generally close to the y = x line until x = 11 min. Above this point the model begins to lose predictive power, and the values begin to fall away from the diagonal. Where applicable, both the x and y axis are reported in terms of the mathematically transformed values.

Click here for additional data file.

Supplemental Information 2 Less well performing models

As in Figure S1, the scatter graphs demonstrate the predictive power of the models. In comparison, these models do not perform as well and therefore have lower predictive power.

Click here for additional data file.

Supplemental Information 3 Heatmap of the Pearson correlation coefficient between the Fv sequence composition scores and the score on each of the experimental datasets for the mAb137 dataset as in Fig. 3

In comparison, only features selected for by each algorithm are chosen to elucidate the relationship between the features selected, and the correlation coefficient. Of note, features associated with charge and hydrophobicity are prominent both for their generally high correlation values, but also the regularity at which they are chosen by each algorithm. Dark red values indicate a stronger positive correlation, and dark blue values indicate a stronger negative correlation.

Click here for additional data file.

Supplemental Information 4 HIC experimental results for each protein in the mAb137 dataset in context of the aromatic content and absolute charge for each Fv sequence

In (A), the aromatic content—here defined as the combined composition value for the amino acids F, W and Y - shows a general positive correlation with the HIC experimental score for that sequence. Sequences with an above average absolute charge value, in the context of the mAb137 dataset, are coloured red, and those with less yellow. In (B), there is a general negative correlation between the absolute charge—calculated from the number of charged residues D, E, K and R within the sequence—and HIC experimental score. Sequences with an above average aromatic content are coloured pink, and those with a below average aromatic content are coloured blue.

Click here for additional data file.

The authors would like to acknowledge feedback from members of the Warwicker and Curtis groups at the University of Manchester.

Additional Information and Declarations

Competing Interests

Author Contributions

Data Availability

The authors declare there are no competing interests.

Max Hebditch conceived and designed the experiments, performed the experiments, analyzed the data, contributed reagents/materials/analysis tools, prepared figures and/or tables, authored or reviewed drafts of the paper, approved the final draft.

Jim Warwicker conceived and designed the experiments, analyzed the data, contributed reagents/materials/analysis tools, authored or reviewed drafts of the paper, approved the final draft.

The following information was supplied regarding data availability:

Data is available at https://github.com/maxhebditch/abpred.

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
