# Peer review of "Charge and hydrophobicity are key features in sequence-trained machine learning models for predicting the biophysical properties of clinical-stage antibodies"

_PeerJ, doi:10.7717/peerj.8199_

## Round 0.1 · original submission · Major Revisions

Please thoroughly revise your manuscript following the reviewers' comments. Also, please improve the clarity of the manuscript and briefly define and explain concepts and quantities (e.g. R^2, VIF,...)

Reviewer 1 ·

Basic reporting

Line 79: The sentence that starts with ‘Clustering of…’ is unclear. Please rephrase it.
Line 136: Give a reference for ‘fitting to a priori approximation’.
Line 147: Give references for validation approaches.

Experimental design

Line 40: A reference for ‘excipient effect’ would be welcome here.
Line 97: Please explain the rationale behind the composite scores.
Line 108: Why did authors assume linearity in their models? What’s the rationale behind?
Line 142: Why 11 models were chosen, which were the selection criteria?
Line 229: How were the sequences concatenated? How did the authors deal with unequal sequence length? How does this translate into processing of an input for your online tool?
Figure 3: How was the heatmap constructed? Are we looking at plain Pearson-correlation coefficients?

Validity of the findings

Line 48: Authors should recognize here semi-empirical tools like mAb Rosetta, which benefit from data, but also theoretical calculations and optimization.
Figure 2: The HIC data, disclosed in Figure 2, depict a non-linear, 2-state like model.
Line 80: The novelty of authors approach is not clear.
Line 238: Can a correlation of R^2<0.5 be regarded as an accurate prediction?
Line 240: Could authors compute the correlation coefficient for higher-HIC values sample?
Line 263: How do authors know the charge of CDRs? Do BVP contain negatively charged proteins, or perhaps vast amounts of negatively charged phospholipids?
Figure 4: How did authors compute the ‘absolute charge’ of a protein?
Line 305: How did authors arrive at this conclusion? What's the evidence supporting the intermolecular repulsion vs retention hypothesis. Have authors analyzed SINS-type of assays. Is there a correlation with these?
Line 349: The conclusion that ‘linear models can account for much of the variation’ is simply contradictory to the finding reported in Line 292.

Additional comments

I would like to commend the authors for uptaking a truly daunting task of building sequence-based models for mAb developability assessment. The online tool presented by the authors holds a promise of accelerating the rational design of therapeutic mAbs.

However, I do believe the manuscript is not suitable for publication in the current form.

The word ‘charge’ appears throughout the manuscript 42 times. However, it is evident that authors refer here to a relative number of titratable residues. Thus the manuscript needs to be rewritten to make it clear that no charge-charge calculations are done and that intrinsic biophysical properties of amino acids only are used in the design of the study.

The title of a manuscript implies that the said machine learning methodologies can be used in the prediction of biophysical properties, yet what is predicted are ‘normalized’ values of developability screening techniques, which are de facto distant proxies of biophysical properties, such as, self-aggregation or intrinsic solubility.

The study disclosed in the manuscript suffers greatly from a lack of description of the model selection process and (it seems) an arbitrary selection of ML methodologies. The study demands a question, why would one choose a specific ML methodology? Is there an added benefit, e.g. the ability of the method to capture a biophysical property that goes beyond an optimization against a simple R^2 metric?

The conclusions for HIC calculations are contradictory to the main takeaway of the results section: the HIC profile is not linear, in fact displays a two-state-like behavior, thus it begs to be fitted with (at least) two independent lines.

Finally, the manuscript does not explain the relative benefit of authors’ approach with respect to the existing sequence-based prediction methods. Could the existing tools be successfully used to arrive at the conclusions of this study?

Reviewer 2 ·

Basic reporting

no comment

Experimental design

no comment

Validity of the findings

no comment

Additional comments

I think this is a potentially interesting contribution on understanding of antibody biophysical properties and the web-server provided is functional and easy to use.

1. My main concern is the size of the dataset. There are only 137 proteins but 35 features are used which is very easy to get over-fitting using machine learning approaches. Is it possible to collect more proteins with experimental data from other papers?
2. Are there any similar methods available to compare the prediction performance?
3. In Uemura et al (2018), it is reported that the molecular size and the solubility are negatively correlated. Could you also check the correlation between the protein size and biophysical properties? If the correlation is high, it might be used as one feature.

Could you also show the correlation between the experimental values and the most important features, e.g., charge and hydrophobicity?

4. For some datasets, for example, HEK and DSF, the machine learning models give worse predictions. Could you explain why?
5. I think the definition of ‘biophysical properties’ need to be better explained in the introduction. Which biophysical properties?
6. The figures in Supplement files are hard to read and need a higher resolution.

---

## Round 0.2 · accepted · Accept

The revised manuscript contains all the needed improvements.